# Inversion modeling of japonica rice canopy chlorophyll content with UAV hyperspectral remote sensing

Yingli Cao[1,2], Kailun Jiang[1,2], Jingxian Wu[3], Fenghua Yu[1,2], Wen Du[1,2], Tongyu Xu[1,2]*

**1** Department of Information and Electrical Engineering, Shenyang Agricultural University, Shenyang, China,
**2** Liaoning Engineering Research Center for Information Technology in Agriculture, Shenyang, China,
**3** Department of Electrical Engineering, University of Arkansas, Fayetteville, AR, United States of America

* yatongmu@163.com

**Data Availability Statement:** All data files are available from https://github.com/Yingli-C-SYAU/Related-programs-and-code.

**Funding:** The work for YC and JK was supported by National Key R&D Program of China

## Abstract

Chlorophyll content is an important indicator of the growth status of japonica rice. The objective of this paper is to develop an inversion model that can predict japonica rice chlorophyll content by using hyperspectral image of rice canopy collected with unmanned aerial vehicle (UAV). UAV-based hyperspectral remote sensing can provide timely and cost-effective monitoring of chlorophyll content over a large region. The study was based on hyperspectral data collected at the Shenyang Agricultural College Academician Japonica Rice Experimental Base in 2018 and 2019. In order to extract the salient information embedded in the high-dimensional hyperspectral data, we first perform dimension reduction by using a successive projection algorithm (SPA). The SPA extracts the characteristic hyperspectral bands that are used as input to the inversion model. The characteristic bands extracted by SPA are 410 nm, 481 nm, 533 nm, 702 nm, and 798 nm, respectively. The inversion model is developed by using an extreme learning machine (ELM), the parameters of which are optimized by using particle swarm optimization (PSO). The PSO-ELM algorithm can accurately model the nonlinear relationship between hyperspectral data and chlorophyll content. The model achieves a coefficient of determination $R^2 = 0.791$ and a root mean square error of RMSE = 8.215 mg/L. The model exhibits good predictive ability and can provide data support and model reference for research on nutrient diagnosis of japonica rice.

## Introduction

Japonica rice is a major Asian rice variety with a large cultivation area in Northeastern China. Japonica rice has a long growth period, high protein content, and good flavor [1]. Chlorophyll content is an important indicator for characterizing the growth status of japonica rice. Non-destructive, rapid, and large-scale estimation of the chlorophyll content of japonica rice has long been an important research direction for precision agriculture [2–5]. Recently there have been growing interests of employing unmanned aerial vehicle (UAV) low-altitude remote sensing platforms in precision agriculture. Compared to other remote sensing methods, UAV remote sensing platforms have the advantages of low cost, high flexibility, large coverage area, and versatility in

(2017YFD0300700). The work of JW was supported in part by the U.S. National Science Foundation (NSF) under Award Number ECCS-1711087.

**Competing interests:** The authors have declared that no competing interests exist.

terms of collected data. This motivates the development of UAV hyperspectral remote sensing platform for rapid assessment of japonica rice chlorophyll content at near-earth scale, which has important practical significance for assisting field precision fertilization and pesticide application.

Accurate, rapid and nondestructive access to rice growth information is essential for timely evaluation of rice growth status, which can provide valuable information for crop fertilization. Traditional methods of measuring crop growth indicators, such as chlorophyll content, leaf area index, biomass, etc., are mainly manual measurements with indoor chemical analysis, which is time-consuming and labor intensive. The manual approach is often poorly timed, and it seriously limits the timeliness and effectiveness of crop growth monitoring. In addition, most existing studies employ conventional optical sensors, which cover only the visible light spectrum. On the other hand, hyperspectral sensors cover both the visible light spectrum and part of the infrared spectrum, and it can obtain sensitive information that are critical to rice growth status evaluation. Therefore, the development of hyperspectral monitoring system on UAV platforms can significantly improve the efficiency and accuracy of growth status monitoring and precision management of rice field.

Hyperspectral data have been used to assess chlorophyll content for a variety of crops [6–9]. In [6], hyperspectral imaging of potato leaves were collected at different vertical leaf positions, and a rational function-partial least squares (RF-PLS) model was developed to estimate the chlorophyll content at different vertical leaf positions by using the hyperspectral data. The results show that the hyperspectral reflectance varies with different vertical leaf positions. In [7], Li et al. studied the reflectance spectrum of winter wheat canopy in the range of 325 to 1075 nm, and a genetic algorithm (GA) was used to optimize the spectral characteristic parameters. In combination with correlation analysis, a least squares-support vector regression method was used to establish a winter wheat chlorophyll content prediction model, which provided support for subsequent fertilization decisions. A UAV low-altitude remote sensing platform was used to collect hyperspectral images of wheat canopy in [8]. A leaf chlorophyll content inversion model was developed by establishing a comprehensive growth index (CGI) during a key wheat production period. The inversion model achieved a coefficient of determination greater than 0.7, which provides a good reference for monitoring wheat chlorophyll content. In [9], hyperspectral data were used to estimate the canopy cholorophyll content of spring wheat by using a partial least squares regression model. It was found that integrating the maximum reflectance of the hyperspectral data from 820 to 940 nm can provide an accurate estimate of the chlorophyll content.

Most current research on crop chlorophyll inversion with hyperspectral data were developed by using statistical regression models, and they can be roughly divided into two categories: vegetation index models [10–12], and direct spectrum models [13, 14]. In the vegetation index models, the hyperspectral data are first used to construct various vegetation indices, which are then used to develop multiple linear or nonlinear regression methods to establish an inversion model between these indices and chlorophyll content. Such an approach has the advantage of simple model constructions with clear physical meanings. However, it requires the construction of a large number of spectral indices in an ad hoc manner, without a systematic guidance to general crop varieties. The various spectral indices depend on the type and region of a given species, thus it lacks universal applicability [10–12]. The direct spectrum models rely on the modeling of the entire hyperspectral bands, which is usually a high dimension vector. Direct utilizing the entire hyperspectral band can results in high model complexity or even model overfitting [13]. This problem can be partially solved by employing dimension reduction techniques, such as principal component analysis (PCA) or PLS [14].

The hyperspectral data of most of the above existing works were collected on the ground or at low altitude. Therefore the coverage area of the measurement is very small. On the other

hand, the proposed UAV platform can operate at a higher altitude, which enables efficient large scale data collection. Specifically, the proposed system operates at an altitude of 150 m, and can obtain the hyperspectral image of an 1000 $m^2$ area in 15 seconds. In addition, many existing methods in the literature were developed by using traditional regression methods, where many of the model parameters were initialized in a heuristic manner based on past experience. The proposed approach using a data-drive approach, such that all model parameters are obtained through the collected data. The data-driven approach can remove human bias, and obtain a more accurate prediction results.

The objective of this work is to develop an inversion model that can predict japonica rice canopy chlorophyll content by using hyperspectral imaging (HSI) data collected with a UAV remote sensing platform. The pixels in the HSI image are obtained by using hyperspectral reflectance data, that is, each pixel in the image contains the hyperspectral reflectance information at the corresponding location. The study was based on data collected at the Shenyang Agricultural College Academician Japonica Rice Experimental Base in 2018 and 2019. In addition to hyperspectral data of japonica rice canopy collected by UAVs, rice samples were also collected on the ground and analyzed in the lab to provide ground truth for the inversion model. In order to reduce model complexity and avoid model overfitting, we first employ a successive projection algorithm (SPA) to extract the characteristic bands from the high dimension hyperspectral vector. Based on our experiment data, the characteristic bands extracted by SPA are 410 nm, 481 nm, 533 nm, 702 nm, and 798 nm, respectively. With the extracted characteristic bands as input, the inversion model is developed by using an extreme learning machine (ELM) [15]. The parameters of ELM, including population size, inertia weight, learning factor, and the velocity position correlation coefficient, are optimized by using particle swarm optimization (PSO) [16].

ELM is a feed-forward neural network with a single or multiple hidden layers. Unlike conventional neural network with back propagation (BP), the parameters of the nodes in the hidden layers of ELM are randomly assigned and never tuned. One main advantage of ELM is that it can learn much faster compared to conventional neural network with BP. However, due to the randomness of the parameters of the hidden layer nodes, a large number of hidden layer nodes are usually needed to achieve the desired accuracy in many practical applications. In addition, the conventional ELM architecture sometimes does not have good generalization capability [17, 18]. To address the above issues, we propose to employ a particle swarm limiting learning machine algorithm that employs a combination of PSO and ELM [19]. Specifically, the PSO algorithm is used to optimize the weights of the input layer and the deviations of the hidden layer. In this case, the number of hidden layer nodes is learned automatically from the data, such that a good optimization performance can be achieved with a relatively small number of hidden layer nodes with low complexity.

It has been demonstrated in this paper that the PSO-ELM algorithm outperforms conventional ELM algorithm in terms of the number of hidden layer nodes and network generalizations. The PSO-ELM algorithm accurately models the nonlinear relationship between hyperspectral data and chlorophyll content. The proposed UAV-HSI platform equipped with the PSO-ELM algorithm can facilitate the development of precision agriculture system, increase crop yields, and promote the socio-economic development of the society.

## Materials and methods

### Experiment setup

The experiment was carried out between June and September in 2018 and 2019, respectively. It took place at the Liaozhong Kalima Academician Rice Experimental Station (41˚47′N,122˚

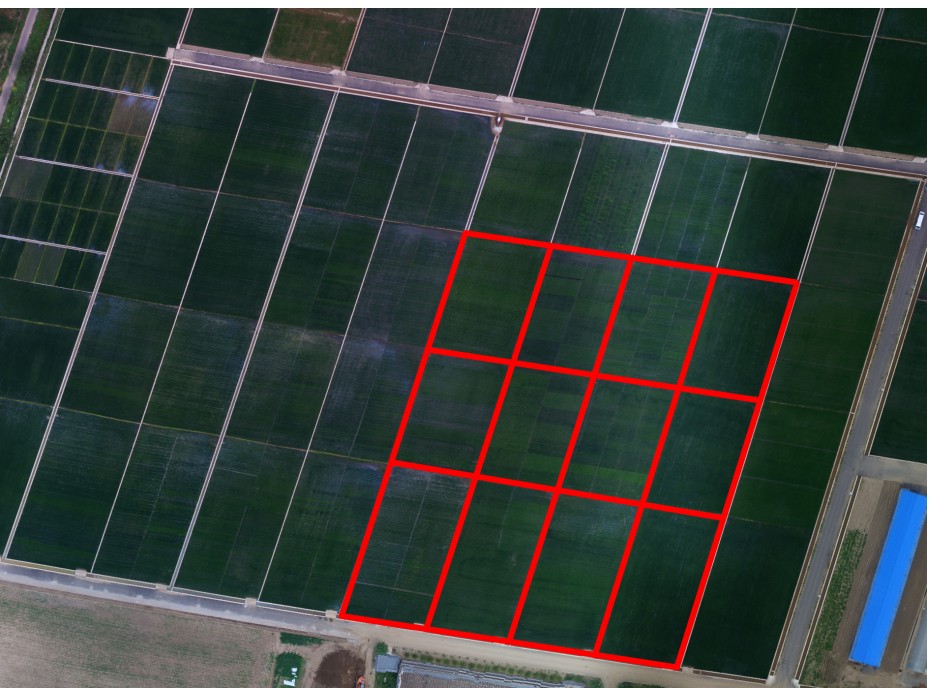

**Fig 1. Test plots used for data collection.**

71′E) of the Shenyang Agricultural University. The rice strain was "Shendao 529"" The japonica rice test plots were designed with four different nitrogen fertilizer gradient treatments, namely CK, $N_1$, $N_2$ and $N_3$. Each treatment was repeated three times. There were a total of 12 test plots, and the size of each test plot is $30 \times 45\ m^2$. Among them, CK represents the control group, i.e., no nitrogen fertilizer was applied; $N_1$ represents the local standard nitrogen fertilizer application level, and the nitrogen fertilizer application amount was 45 kg/ha; $N_2$ represents the low nitrogen fertilizer application level, and the application amount was half that of $N_1$; $N_3$ is the group with the high nitrogen fertilization level, and the application amount was 1.5 times that of $N_1$. The applications of phosphate fertilizer and potash fertilizer were carried out based on local standard application quantities, that is, 51.75 kg/ha for phosphate fertilizer, and the standard application amount is 18 kg/ha for potassium fertilizer. Fig 1 shows the design of the test area and test plots for this study.

## UAV hyperspectral image data acquisition

The UAV hyperspectral platform adopts the M600 PRO six-rotor UAV from Shenzhen Dajiang Innovation Co., Ltd. (Shenzhen, Guangdong, China). The hyperspectral images were collected by using the GaiaSky-mini built-in push-broom airborne hyperspectral imaging system from Sichuan Shuangli Hepu Company (Sichuan, China). The frequency range of the data acquired by the hyperspectral imaging sensor is 400 to 1000 nm. The data is preprocessed by using spectral difference calculations with a hyperspectral resampling interval set to 2.35 nm, which results in 255 bands over the spectral range between 400 to 1000 nm. The two bands on the upper and lower boundaries of the spectrum are removed, which leads to 253 effective bands. The hyperspectral imaging system is shown in Fig 2.

The Gaiasky-mini airborne hyperspectral imaging system is used to obtain hyperspectral remote sensing images of the rice canopy. The hyperspectral imaging sensor has a built-in

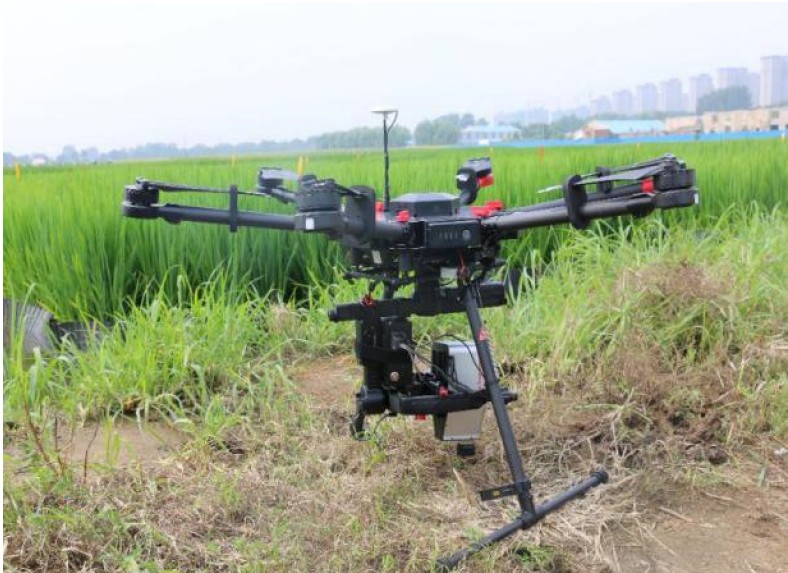

**Fig 2. UAV hyperspectral imaging system.**

push-sweep imaging function, thus the hyperspectral image was obtained by keeping the drone hovering steadily over the rice fields during the data acquisition process. During data acquisition, the drone is hovered with an altitude of 150 m and stationary to the ground. and the corresponding coverage area of one hyperspectral scene image is 1000 $m^2$. The acquisition time of a scene image is 15 seconds. All hyperspectral images used in this work are obtained during the rice fertility period.

The acquisitions of the hyperspectral images were performed on selected Wednesdays between 11:00 AM and 12:00 PM in the morning. The data were collected on days with sunny or lightly cloudy weather (cloud less than 20%), and the solar light intensity was relatively stable. The flying height of the UAV was a true height of 150 m. The collected hyperspectral data were processed and extracted with the ENVI5.3+IDL software tool. During processing, the spectral angle mapper (SAM) method was first used to remove impacts from interfering objects, then the hyperspectral image was generated by calculating the average spectrum of each area of interest. Before data analysis, hyperspectral reflectance calibration was performed to achieve black-white correction in the HSI image. The hypersectral reflectance calibration is performed as follows:

$$\rho_t = \frac{DN_t - DN_1}{DN_2 - DN_1}(\rho_2 - \rho_1) + \rho_1$$

where $\rho_t$ and $DN_t$ are, respectively, the reflectivity and DN values of the ground objects to be converted, $DN_1$ and $DN_2$ are the DN values of the calibration blanket, and $\rho_1$ and $\rho_2$ are the reflectivity of the calibration blanket.

## Ground data acquisition and analysis

In addition to hyperspectral image data, we also performed manual ground data collection in the japonica rice field. Data collected on the ground will serve as ground truth for the UAV hyperspectral data. A total of 196 rice plant samples were collected and analyzed for ground data acquisition and analysis, with 4 plants in each test plot. The four rice cavities selected for

the same sample for this study were based on the similarity of rice growth in the rice experimental area. The sampled rice plants were located in the middle of the test plots where the growth was even. The 196 ground data samples cover all six stages of the key growth periods of japonica rice, including tillering, jointing, booting, heading, flowering, and filling. Every 50 canopy leaves were placed in a sealed bag and stored in a cooler with temperature around 4°C. A hand-held differential GPS device (manufacture: XAG, model: RTK, precision: cm scale) was used to measure the geographical coordinates of the sampling points, which were then correlated with the hyperspectral data collected by the UAV [22–22]. The hyperspectral remote sensing image obtained by UAV hyperspectral remote sensing platform contains GPS information of each pixel. The ground data of a given sample are then mapped to a pixel in the HSI data with the same GPS coordinate.

After the japonica rice samples were returned to the laboratory, fully-expanded leaves of the japonica rice samples were selected and cut into small pieces. Pieces from the same leaf are mixed together, and 0.4 g of the leaf mixture was placed in 200 mL of an extraction solution, which contains acetone, ethanol, and distilled water at a ratio of 9:9:2. The solutions with leaf mixture are left to stand in the shady area of a laboratry with temperature around 20°C. Once the leaf sample appeared to be completely white, colorimetry was then performed with a Hengping 754 UV-visible spectralphotometer. The spectralphotometer covers the wavelength range from 190 nm to 1000 nm, with a step size of 0.1 nm. The spectrophotometer is to measure the optical density (OD) at 663 nm and 645 nm, respectively. The principle of spectrophotometer measurement is to measure the absorption intensity of chlorophyll solution at different wavelengths, and the strength of chlorophyll content can then be deduced from the absorption intensity at different wavelengths. For spectroscopic measurement, most of the essential information is expressed at the characteristic wavelengths of 663 nm and 645 nm. It should be noted that the proposed UAV-HSI system is measuring the reflectance information of leaves in solid form, thus the reflectance characteristic wavelengths used by the UAV-HSI platform will be different from 663 nm and 645 nm. Based on the colorimetry result, the chlorophyll content of the japonica rice samples was calculated according to Eq (1) [23]

$$\texttt{Chl(mg/L)} = 5.134 OD_{663} + 20.436 OD_{645} \tag{1}$$

where Chl is the chlorophyll level, and $OD_{663}$ and $OD_{645}$ are the ODs at 663 nm and 645 nm, respectively.

A total of 10 experimental data collections were conducted in this study, and 196 valid data samples were collected. Of these, 84% were selected to form a modeling dataset (165 samples), and the remaining 16% were used to construct a validation dataset (31 samples). Table 1 shows division of the experimental data training set and validation set for this paper. The statistical characteristics of the sampled data are shown in Table 1.

According to Table 1, the modeling and validation dataset share similar statistical properties with the exception of sample size. Additionally, the coefficient of variation was greater than 40%, indicating that the chlorophyll content data exhibited high dispersion.

## Extraction of characteristic hyperspectral bands

The successive projection algorithm (SPA) was used to extract hyperspectral information from characteristic spectral bands of the japonica rice canopy. Information obtained from SPA was then used as input to the chlorophyll content inversion model. Previous research by our group verified that the change in chlorophyll content of japonica rice mainly affects the spectral

**Table 1. Statistical characteristics of chlorophyll content in japonica rice in test plots.**

| Sample set | Number of samples | Chlorophyll content maximum (mg/L) | Chlorophyll content minimum (mg/L) | Chlorophyll content average (mg/L) | Standard deviation (mg/L) | Coefficient of variation (%) |
|---|---|---|---|---|---|---|
| Overall | 196 | 99.70 | 2.60 | 54.66 | 26.94 | 49.28 |
| Modeling dataset | 165 | 99.70 | 2.60 | 54.33 | 27.31 | 50.27 |
| Validation dataset | 31 | 93.80 | 10.70 | 56.42 | 25.22 | 44.70 |

reflectance between 400 to 800 nm Fig 3. The detailed analysis method is described in [24, 25]. Therefore, 400 to 800 nm was selected as the hyperspectral interval in this study.

SPA is a forward variable selection algorithm that minimizes collinearity in a vector space. Define a size $M \times J$ calibration data matrix as $\mathbf{X}_{\text{cal}} = [\mathbf{x}_1,...,\mathbf{x}_J]$, where $J$ is the number of candidate wavelengths, $M$ is the number of calibration waveforms from the hyperspectral image, and $\mathbf{x}_j$ is a length $M$ vector. The objective of SPA is to identify a subset of $N < J$ columns of $\mathbf{X}_{\text{cal}}$ with minimum collinearity, where $N$ is a predefined parameter. Such an approach can identify a subset of wavelengths with minimum redundant information.

SPA performs forward subset selection in an iterative manner. It starts with one wavelength, and then add a new wavelength at each iteration until $N$ wavelengths are selected. Define the set of selected wavelengths as $\mathcal{A}$, and the set of unselected wavelengths as $\mathcal{S}$, such that $\mathcal{A} \cup \mathcal{S} = \mathcal{J} = \{1,...,J\}$.

During the iterative selection process, we will project each vector with wavelength in $\mathcal{S}$ onto the subspace spanned by the vectors indexed by the elements in $\mathcal{A}$. The vector with the largest residue after projection contains the most new information that is not represented by the elements in $\mathcal{A}$, thus the corresponding wavelength will be added to $\mathcal{A}$.

The initial values of $\mathcal{A}$ and $\mathcal{S}$ are $\mathcal{A} = \emptyset$ and $\mathcal{S} = \mathcal{J}$, respectively, where $\emptyset$ is the empty set. At the beginning of the iteration, assign $\mathbf{y}_j = \mathbf{x}_j$, for all $j \in \mathcal{S}$. The values of $\mathbf{Y}_j$ will be iteratively updated throughout the iteration process. Denote the wavelength selected at the $k$-th iteration as $a_k$. The corresponding vector with index $a_k$ is represented as $\mathbf{z}_k = \mathbf{y}_{a_k}$. Each iteration contains the following three steps [26, 27].

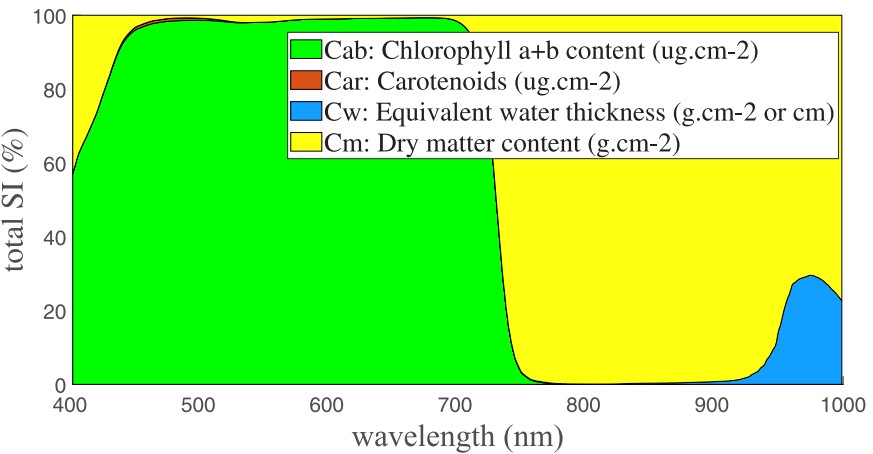

**Fig 3. Global sensitivity analysis of the input parameters in the PROSPECT model.**

Step 1 (Projection). During the $k$-th iteration, we first project all vectors $\mathbf{Y}_j, j \in \mathcal{S}$, into the null space of $\mathbf{z}_{k-1}$ as

$$\mathbf{e}_j = \begin{cases} \mathbf{y}_j - \mathbf{z}_{k-1}(\mathbf{z}_{k-1}^T \mathbf{z}_{k-1})^{-1}\mathbf{z}_{k-1}^T\mathbf{y}_j, & k > 1, \\ \mathbf{y}_j, & k = 1, \end{cases} \quad \forall j \in \mathcal{S} \tag{2}$$

For the first iteration with $k = 1$, since $\mathcal{A}$ is empty, the projection operator to the null space of the vectors indexed by $\mathcal{A}$ is simply the size $M$ identity matrix $\mathbf{I}_M$.

Step 2 (Selection). The index of the residual vector $\mathbf{e}_j$ with the largest $\ell_2$ norm is selected as the new wavelength as

$$a_k = \underset{j \in \mathcal{S}}{\operatorname{argmax}} \quad \|\mathbf{e}_j\|_2 \tag{3}$$

Step 3 (Update). Update the sets $\mathcal{A} = \mathcal{A} \bigcup \{a_k\}$ and $\mathcal{S} = \mathcal{S} \setminus \{a_k\}$, where the operator $\setminus$ represents the difference between two sets. Update all $\mathbf{Y}_j$ as the residual of the projection as

$$\mathbf{y}_j = \mathbf{e}_j, \text{ for all } j \in \mathcal{S}. \tag{4}$$

The output of SPA is the set of wavelength indices $\mathcal{A}$. To further reduce the number of wavelengths, multiple linear regression (MLR) models were applied to all subsets of $\mathcal{A}$. The subset that results in the smallest root mean square error (RMSE) is selected as effective characteristic hyperspectral bands.

## PSO-ELM inversion modeling of chlorophyll content

The inversion model of japonica rice canopy chlorophyll content was established by using the subset of characteristic hyperspectral bands obtained from the previous section. The relationship between the chlorophyll content and the characteristic hyperspectral bands were modeled by using extreme learning machine (ELM) with particle swarm optimization (PSO). The PSO algorithm was used to optimize the selection of input layer weights and hidden layer biases of the ELM to calculate the output weight matrix. The parameters of the PSO algorithm considered in this study mainly include population size (pop), inertia weight (w), learning factor ($c_1$, $c_2$), and the velocity position correlation coefficient (m). It has been demonstrated that the PSO-ELM outperforms conventional ELM algorithm in terms of the number of hidden layer nodes and network generalizations [28]. In this study, the accuracy of the inversion model is assessed by using the RMSE and the coefficient of determination ($R^2$).

## Results and discussion

### Extraction of characteristic hyperspectral band

We first study the correlation between the japonica rice chlorophyll content and the various hyperspectral bands from the japonica rice canopy reflectance data. Three methods, including Pearson's correlation coefficient, distance correlation coefficient, and maximum information coefficient (MIC), were used to analyze the correlation between chlorophyll and each individual band. Before applying the correlation methods, the Savitzky-Golay convolution smoothing algorithm was used to smooth the 400 to 800 nm reflectance spectral data, obtained from the test plots and reduce the effect of noise on chlorophyll content inversion accuracy. The japonica rice canopy hyperspectral curve after noise reduction is shown in Fig 4.

Results from the correlation analysis are shown in Fig 5. Results from all three methods in general have similar trends: the maximum correlation between chlorophyll content and

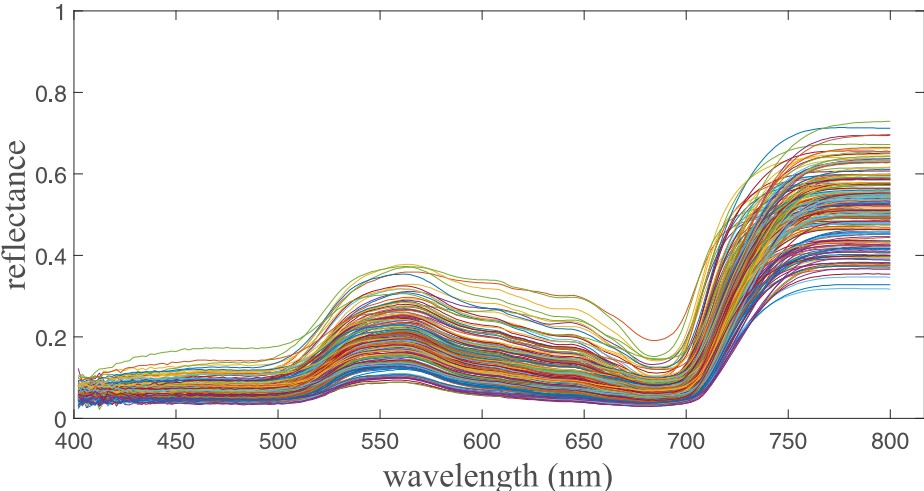

**Fig 4. Noise reduction of hyperspectral information smoothing.**

hyperspectral reflectance is achieved in the wavelength range between 701 to 705 nm. Specifically, the peak correlation is located at the wavelength of 702 nm for all three methods, with the distance, Pearson and MIC correlation coefficients being 0.66, 0.64, and 0.53, respectively. In addition to the maximum correlation band, relatively strong correlations are also observed at other bands, such as the band with wavelength between 410 to 480 nm. Even though the reflectivity of 400 to 450 nm is lower than 0.1, the signal-to-noise ratio of all bands is between 6.0 dB to 15.9 dB, which is sufficient to extract the characteristic information of the response. The SNR of each band is obtained by calculating the ratio between the total power of the signal and the variance of the signal in the range between 400 to 800 nm.

The SPA algorithm described in Section 1 was used to extract characteristic hyperspectral bands for the chlorophyll content inversion model. Based on our experiment data, the extracted characteristic wavelengths were 410 nm, 481 nm, 533 nm, 702 nm, and 798 nm. Fig 6 shows the average reflectance marked with the extracted characteristic bands.

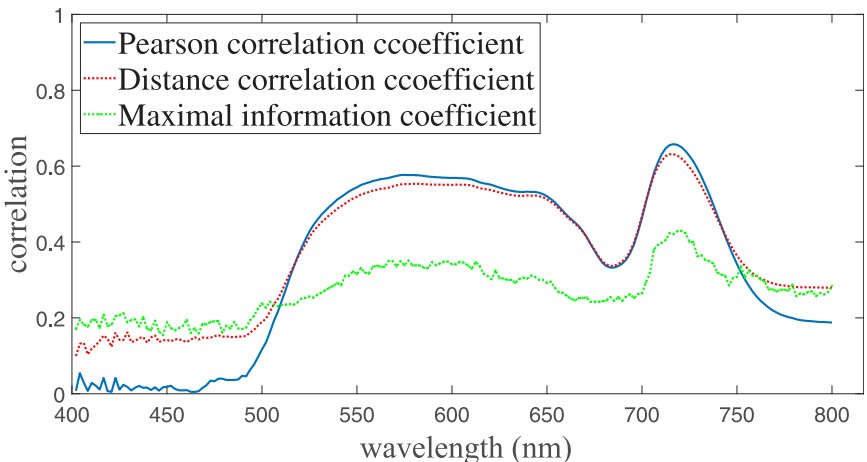

**Fig 5. Single band correlation analysis results.**

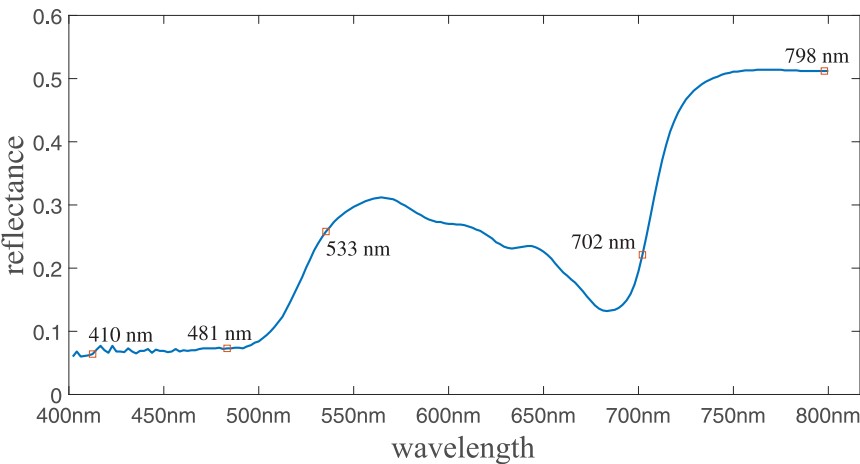

**Fig 6. Selection of characteristic bands of SPA in the canopy layer of japonica rice.**

## PSO-ELM chlorophyll content results

The inversion model of cholorophyll content is studied in this subsection. In the inversion model, the reflectance data on the characteristic hyperspectral bands extracted from SPA are used as independent variables. The chlorophyll contents obtained through ground data acquisition constitute the dependent variables. The inversion model was established by using PSO-ELM. For comparison, we also include the results obtained from conventional ELM. In PSO-ELM, the number of particle swarm iterations was 100. The chlorophyll contents predicted by hyperspectral data with the inversion model are compared to those obtained from ground data, and the performance is evaluated by using the coefficient of determination ($R^2$).

Due to the large number of parameters of the PSO algorithm, an orthogonal array test was designed to identify the optimum PSO parameter settings. The results are shown in Table 2, where each parameter can take one of 5 values, and there are 5 parameters.

Table 3 summarizes the orthogonal array test results of the PSO-ELM parameters, where $W_{ij}$ is the sum of $R^2$ obtained from the $i$-th value of parameter $j$. For example, $W_{11}$ is obtained by calculating the sum of $R^2$ corresponding to the first value of the first parameter, $\texttt{pop} = 40$, and $W_{12}$ is obtained by calculating the sum of $R^2$ corresponding to the first value of the second parameter, $\texttt{w} = 0.3$, as

$$W_{11} \quad = R_1^2 + R_2^2 + R_3^2 + R_4^2 + R_5^2 \tag{5}$$

$$W_{12} \quad = R_1^2 + R_6^2 + R_{11}^2 + R_{16}^2 + R_{21}^2 \tag{6}$$

Other values of $W_{ij}$ are calculated in a similar manner, and the results are shown in Table 3. Each column of Table 3 indicates the impacts of one parameter on the perdiction accuracy. For example, on the first column, the differences among $W_{i1}$, for $i = 1, \ldots, 5$, reveal how the choice of population size affect the accuracy of the inversion model, regardless the values of other parameters. Since $W_{21} \geq W_{i1}$, it is concluded that the population size that maximize $R^2$ shoud be the 2th value in the orthogonal array test, that is, $\texttt{pop} = 50$. Similary, the columns corresponding to $\texttt{w}$, $\texttt{c}^1$, $\texttt{c}^2$, and $\texttt{m}$ are maximized at $W_{52}$, $W_{33}$, $W_{34}$, and $W_{35}$, respectively. Consequently, the optimum values of the parameters are $\texttt{pop} = 50$, $\texttt{w} = 0.9 \sim 0.3$, $\texttt{c}^1 = 1.65$, $\texttt{c}^2 = 2.8$, and $\texttt{m} = 0.2$.

**Table 2. Orthogonal array test for PSO-ELM.**

| No | Model | pop | w | $C_1$ | $C_2$ | m | $C^2$ |
|----|-------|-----|---|-------|-------|---|-------|
| 1 | PSO-ELM | 40 | 0.3 | 1.1 | 1.1 | 0.02 | 0.721 |
| 2 | PSO-ELM | 40 | 0.9 | 1.3 | 1.3 | 0.06 | 0.763 |
| 3 | PSO-ELM | 40 | 1.5 | 1.65 | 1.65 | 0.2 | 0.753 |
| 4 | PSO-ELM | 40 | 0.9~0.3 | 2.8 | 2.8 | 0.6 | 0.773 |
| 5 | PSO-ELM | 40 | 0.3~1.5 | 3.5 | 3.5 | 1 | 0.772 |
| 6 | PSO-ELM | 50 | 0.3 | 1.3 | 1.65 | 0.6 | 0.773 |
| 7 | PSO-ELM | 50 | 0.9 | 1.65 | 2.8 | 1 | 0.758 |
| 8 | PSO-ELM | 50 | 1.5 | 2.8 | 3.5 | 0.02 | 0.759 |
| 9 | PSO-ELM | 50 | 0.9~0.3 | 3.5 | 1.1 | 0.06 | 0.757 |
| 10 | PSO-ELM | 50 | 0.3~1.5 | 1.1 | 1.3 | 0.2 | 0.773 |
| 11 | PSO-ELM | 60 | 0.3 | 1.65 | 3.5 | 0.06 | 0.761 |
| 12 | PSO-ELM | 60 | 0.9 | 2.8 | 1.1 | 0.2 | 0.765 |
| 13 | PSO-ELM | 60 | 1.5 | 3.5 | 1.3 | 0.6 | 0.721 |
| 14 | PSO-ELM | 60 | 0.9~0.3 | 1.1 | 1.65 | 1 | 0.773 |
| 15 | PSO-ELM | 60 | 0.3~1.5 | 1.3 | 2.8 | 0.02 | 0.746 |
| 16 | PSO-ELM | 70 | 0.3 | 2.8 | 1.3 | 1 | 0.739 |
| 17 | PSO-ELM | 70 | 0.9 | 3.5 | 1.65 | 0.02 | 0.779 |
| 18 | PSO-ELM | 70 | 1.5 | 1.1 | 2.8 | 0.06 | 0.687 |
| 19 | PSO-ELM | 70 | 0.9~0.3 | 1.3 | 3.5 | 0.2 | 0.773 |
| 20 | PSO-ELM | 70 | 0.3~1.5 | 1.65 | 1.1 | 0.6 | 0.773 |
| 21 | PSO-ELM | 80 | 0.3 | 3.5 | 2.8 | 0.2 | 0.773 |
| 22 | PSO-ELM | 80 | 0.9 | 1.1 | 3.5 | 0.6 | 0.762 |
| 23 | PSO-ELM | 80 | 1.5 | 1.3 | 1.1 | 1 | 0.733 |
| 24 | PSO-ELM | 80 | 0.9~0.3 | 1.65 | 1.3 | 0.02 | 0.760 |
| 25 | PSO-ELM | 80 | 0.3~1.5 | 2.8 | 1.65 | 0.06 | 0.760 |
| | ELM | | | | | | 0.667 |

With the set of parameters obtained through the orthogonal array test, we can obtain the PSO-ELM japonica rice chlorophyll content inversion model. Applying the newly developed inversion model to the experiment data, we obtain $R^2 = 0.791$ and RMSE = 8.215 mg/L. These results are much better than the conventional ELM model, which obtains $R^2 = 667$ and RMSE = 11.308 mg/L. The inversion effect is shown in Fig 7. From the inversion results, it can be seen that the PSO-ELM used in this study demonstrates significantly better prediction performance for chlorophyll content than the model established by the conventional ELM method.

**Table 3. Analysis of the PSO-ELM orthogonal array test results.**

| | pop | w | $C_1$ | $C_2$ | m |
|---|-----|---|-------|-------|---|
| $W_{1j}$ | 3.782 | 3.767 | 3.716 | 3.749 | 3.765 |
| $W_{2j}$ | **3.820** | 3.827 | 3.788 | 3.756 | 3.728 |
| $W_{3j}$ | 3.766 | 3.653 | **3.805** | **3.838** | **3.837** |
| $W_{4j}$ | 3.751 | 3.824 | 3.796 | 3.737 | 3.802 |
| $W_{5j}$ | 3.788 | **3.836** | 3.802 | 3.827 | 3.775 |

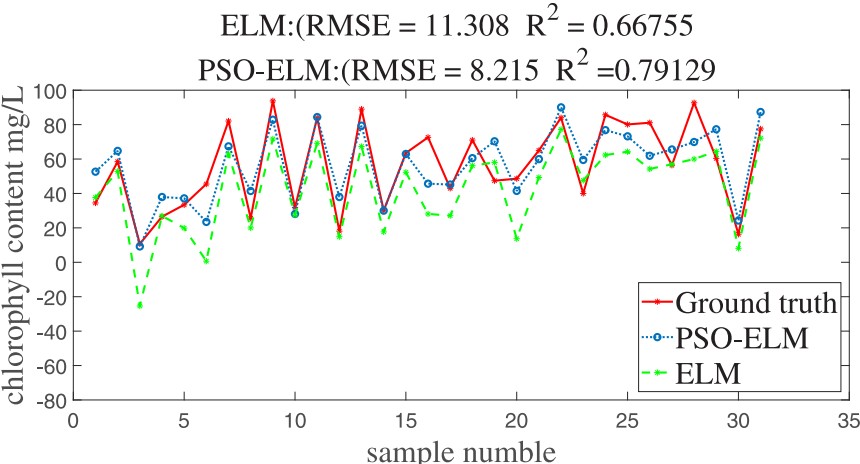

**Fig 7. Inversion results of japonica rice canopy chlorophyll content.**

## Discussions

We have developed an inversion model that can estimate the japonica rice chlorophyll content by using hyperspectral data of japonica rice canopy collected by UAVs. The inversion model enables rapid monitoring and assessment of japonica rice chlorophyll content with UAV remote sensing. Compared to existing methods, the proposed inversion model has two main contributions.

First, an efficient dimension reduction method with SPA was employed to extract the salient information embedded in the high dimensional hyperspectral data. The SPA-based dimension reduction approach provides a more efficient and accurate representation of the hyperspectral contents than the conventional one-dimensional or multiple-regression statistical models, which may not fully express the nonlinear relationship between spectral information and chlorophyll contents [29–32]. The subset of the characteristic hyperspectral band extracted by the SPA algorithm are used as the input of the inversion model.

Second, the inversion model was constructed by using the PSO-ELM algorithm. Conventional ELM algorithm suffers from poor generalization ability and low calibration accuracy due to random input weights and hidden layer thresholds when there are fewer hidden layer nodes. To solve these problems, the PSO algorithm provides a systematic way to optimize input weights and hidden layer thresholds, thus renders a much better model accuracy compared to conventional ELM algorithms.

The five characteristic bands extracted in this study are 410 nm, 481 nm, 533 nm, 702 nm, 798 nm. The two bands in 702 nm and 798 nm are are consistent with the 705 nm and 750 nm range used for chlorophyll content retrieval index in [33]. The characteristic bands of 410 nm, 481 nm, and 533 nm in this paper are similar to those bands used in the study of the rice jointing stage in the literature [34]. Comprehensive analysis shows that there are overlaps and differences in the spectral regions of the characteristic bands among different studies. The differences in characteristic bands are mainly caused by differences in strain varieties, growth periods, environmental conditions, and data processing methods. The difference in the specific characteristic bands is mainly due to the following factors: 1) the rice variety used in this study is different from previous works; and 2) the hyperspectral information might be affected by the presence of water, weeds, and soil in rice fields.

The accuracy of the inversion model can be improved by increasing the amount of data collected from both the UAV platform and ground data. More data can reduce the uncertainties caused by noise, interference, and inevitable collection errors. In addition, the inversion model was established only for experimental japonica rice varieties. Future research will focus on increasing the number of test species and establishing chlorophyll content inversion models for different growth stages of japonica rice in order to improve model accuracy and universal applicability.

The proposed UAV-HSI platform enables rapid, accurate, and non-destructive assessment of rice growth information. The method developed in this paper enables us to obtain timely understanding of the rice growth status, which is essential for rice fertilization planning.

One challenge faced by the proposed method is that the chlorophyll content of rice leaves varies with changes in rice varieties, fertility period, growing environment and other factors. The results obtained in this work are applicable to the crop stage of northeastern rice. New model parameters will need to be obtained for other rice strains.

## Conclusions

We have developed an inversion model that can predict japonica rice chlorophyll content by using rice canopy hyperspectral data collected from UAV platforms. The model was developed by using data collected from the Shendao 529 rice variety with both UAV and ground measurement performed in 2016 and 2017. The inversion model was developed by using a combination of the SPA method and the PSO-ELM algorithm. Specifically, the SPA method was used to achieve hyperspectral data dimension reduction by extracting the hyperspectral characteristic bands, which were then used used as the input to the PSO-optimized ELM algorithm to establish the nonlinear relationship between the chlorophyll content and hyperspectral data. Based on the experimental results, we have the following conclusions.

- The SPA algorithm is effective in extracting hyperspectral characteristic bands in the range of 400 to 800 nm. The characteristic hypersepctral bands selected by the SPA algorithm were 410 nm, 481 nm, 533 nm, 702 nm, and 798 nm, respectively.

- The proposed inversion model achieves $R^2$ = 0.791 and RMSE = 8.215 mg/L, which are significantly better than the inversion model of the conventional ELM algorithm with $R^2$ = 0.667 and RMSE = 11.308 mg/L. The model exhibits good predictive ability and can provide data support and model reference for Northeast japonica rice chlorophyll content research and nutrient diagnosis.

## Acknowledgments

We greatly thank Dr. Dianrong Ma and Minghui Zhao at Shenyang Agricultural College Academician Japonica Rice Experimental Base for providing experimental fields and cultivating the paddies.

## Author Contributions

**Data curation:** Kailun Jiang, Fenghua Yu.

**Funding acquisition:** Yingli Cao.

**Investigation:** Kailun Jiang.

**Methodology:** Yingli Cao, Jingxian Wu, Fenghua Yu.

**Supervision:** Tongyu Xu.

**Validation:** Wen Du.

**Writing – original draft:** Yingli Cao, Kailun Jiang, Fenghua Yu.

**Writing – review & editing:** Jingxian Wu, Tongyu Xu.

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
