## [Decision Letter · Decision Letter 0]

13 May 2020

PONE-D-20-07987

Inversion Modeling of Japonica Rice Canopy

Chlorophyll Content with UAV Hyperspectral

Remote Sensing

PLOS ONE

Dear Dr Xu,

Thank you for submitting your manuscript to PLOS ONE. After careful consideration, we feel that it has merit but does not fully meet PLOS ONE’s publication criteria as it currently stands. Therefore, we invite you to submit a revised version of the manuscript that addresses the points raised during the review process.

Editor's Comment:

As an international journal, PLOS ONE aims to publish studies in a broad international community which should be reflected in any part of the paper. Thus, in addition to the comments from the reviewers, I suggest the authors update their references in a more balanced review as I saw that most of the cited literatures are published by Chinese authors or Chinese journals. A thorough improvement on the language should also be done before I reconsider if the paper is suitable for publication in PLOS ONE.

We would appreciate receiving your revised manuscript by 27th June. To enhance the reproducibility of your results, we recommend that if applicable you deposit your laboratory protocols in protocols.io, where a protocol can be assigned its own identifier (DOI) such that it can be cited independently in the future. For instructions see: http://journals.plos.org/plosone/s/submission-guidelines#loc-laboratory-protocols

We look forward to receiving your revised manuscript.

Kind regards,

Wang Li

Academic Editor

PLOS ONE

Journal Requirements:

2. Our internal editors have looked over your manuscript and determined that it is within the scope of our Plant Phenomics & Precision Agriculture Call for Papers. This collection of papers is headed by a team of Guest Editors for PLOS ONE. The Collection will encompass a diverse range of research articles spanning disciplines, methods and applications.  Additional information can be found on our announcement page: https://plos.io/phenomics. If you would like your manuscript to be considered for this collection, please let us know in your cover letter and we will ensure that your paper is treated as if you were responding to this call. If you would prefer to remove your manuscript from collection consideration, please specify this in the cover letter.

Additional Editor Comments (if provided):

Reviewers' comments:

Reviewer's Responses to Questions

**Comments to the Author**

1. Is the manuscript technically sound, and do the data support the conclusions?

Reviewer #1: Yes

Reviewer #2: Yes

2. Has the statistical analysis been performed appropriately and rigorously? 

Reviewer #1: Yes

Reviewer #2: Yes

3. Have the authors made all data underlying the findings in their manuscript fully available?

Reviewer #1: Yes

Reviewer #2: Yes

4. Is the manuscript presented in an intelligible fashion and written in standard English?

Reviewer #1: Yes

Reviewer #2: Yes

5. Review Comments to the Author

Reviewer #1: In this work, the possible use of UAV-HSI to predict the Chlorophyll content was evaluated. However, several questions should be explained or improved, including the writing expression, they are listed below:

1. Why use the HSI combined with UVA to detect the chlorophyll content? The introduction of significance of this work should be improved, especially in Section Introduction and Section Discussion.

2. The data processing without any black-white correction is not enough rigorous. Raw data from HSI contains tremendous amount of information including samples and noises. It’s very important to eliminate the interference of the natural lighting (or weather conditions) in this work. So, in my opinion, the system with BW correction can promote the applicability of UAV-HSI.

3. From the figures 4 to 6, the spectral reflectance in the range of 400 and 450 nm was lower than 0.1, which means low SNR (Signal to noise ratio) in this wavelengths, why did the authors still use the data to do further analysis?

4. For the selection of key spectral wavelengths, normally one detect the OD values at 663 nm and 645 nm with spectrophotometer, and then calculate the values with equations to describe the contents of chlorophyll contents, and thus these two wavelengths should be selected as key wavelengths.

5. Large area of the paddy field were selected as regions of interest, why only four japonica rice plants in test plots were used for chemical experiments?

6. The sample division for calculated and predicted datasets should be explained and provided in the manuscript.

7. Method for UAV-HSI data acquisition is unclear.

8. Unit for RMSE should be provided.

9. Why PSO-ELM was employed?

10. It would be better if authors can discuss the actual values and practical challenges of the UVA-HSI in modeling of chlorophyll content prediction.

Reviewer #2: The topic of this work is very interesting for the scientific community in the field of Remote Sensing. The monitoring of rice via remote sensing is of vital importance for food and environmental security in a global context, including growth status, yield predictions, area estimates, agricultural insurance and so on. The chlorophyll content is an important indicator of the growth status of rice. The inversion rice chlorophyll content has already been carried out many times using handheld spectroradiometer. However, in this work, the authors access rice canopy chlorophyll content using hyperspectral imaging sensor with UAV platform which can be more efficient and better synchronization. The goals of the paper are very clear and very interesting, and they are obtained in a better way that can be useful. The manuscript is very well written, and I think that it is worthy to be published in this journal after revised.

However, there are some questions and considerations that I would like the authors explain in the manuscript. I already marked some comment in the pdf document.

6. PLOS authors have the option to publish the peer review history of their article (what does this mean?). If published, this will include your full peer review and any attached files.

Reviewer #1: No

Reviewer #2: No

---

## [Author Response · Author response to Decision Letter 0]

3 Jul 2020

We would like to thank the editor and the reviewers for their careful reviews and high quality review comments, which have significantly helped us revise and improve the quality of the paper. The comments are very constructive and extremely helpful during the revision process. All of the reviewers' comments have been addressed in the revised version. A detailed point-by-point response has been included in a separate document: “Response to Reviewers”. All modifications made in the paper has been highlighted in yellow.

---

## [Decision Letter · Decision Letter 1]

31 Jul 2020

PONE-D-20-07987R1

Inversion Modeling of Japonica Rice Canopy

Chlorophyll Content with UAV Hyperspectral

Remote Sensing

PLOS ONE

Dear Dr. Tongyu Xu,

Thank you for submitting your manuscript to PLOS ONE. After careful consideration, we feel that it has merit but does not fully meet PLOS ONE’s publication criteria as it currently stands. Therefore, we invite you to submit a revised version of the manuscript that addresses the points raised during the review process.

You will see from the below comments that the reviewer 1 still have some questions for you to further improve the paper.

We look forward to receiving your revised manuscript.

Kind regards,

Wang Li

Academic Editor

PLOS ONE

Reviewers' comments:

Reviewer's Responses to Questions

**Comments to the Author**

1. If the authors have adequately addressed your comments raised in a previous round of review and you feel that this manuscript is now acceptable for publication, you may indicate that here to bypass the “Comments to the Author” section, enter your conflict of interest statement in the “Confidential to Editor” section, and submit your "Accept" recommendation.

Reviewer #1: All comments have been addressed

Reviewer #2: All comments have been addressed

2. Is the manuscript technically sound, and do the data support the conclusions?

Reviewer #1: Yes

Reviewer #2: Yes

3. Has the statistical analysis been performed appropriately and rigorously? 

Reviewer #1: Yes

Reviewer #2: Yes

4. Have the authors made all data underlying the findings in their manuscript fully available?

Reviewer #1: Yes

Reviewer #2: Yes

5. Is the manuscript presented in an intelligible fashion and written in standard English?

Reviewer #1: Yes

Reviewer #2: Yes

6. Review Comments to the Author

Reviewer #1: Is it a mistake saying the hyperspectral band range is 400 to 100 nm and with a 3 nm resolution? Why the number of effective bands is 253? Please have a check and make necessary changes (100-1000 nm) .

Although the authors added some references (33-38), it would be better to discuss further or compare your results with the published data.

Reviewer #2: (No Response)

7. PLOS authors have the option to publish the peer review history of their article (what does this mean?). If published, this will include your full peer review and any attached files.

Reviewer #1: No

Reviewer #2: No

---

## [Author Response · Author response to Decision Letter 1]

14 Aug 2020

Response to Comments of Reviewer

Thanks for your careful review and valuable comments for the improvement of this paper. The

authors appreciate your detailed comments and valuable suggestions on the technical contents.

Your technical concerns and suggestions are addressed as follows.

1. Is it a mistake saying the hyperspectral band range is 400 to 100 nm and with a 3 nm resolution? Why the number of effective bands is 253? Please have acheck and make necessary changes (100-1000 nm) 

Response:

Thanks for your question. We have clarified the number of effective bands in Section II-B of the revised paper as follows: The frequency range of the data acquired by the hyperspectral imaging sensor is 400 to 1000 nm. The data is preprocessed by using spectral difference calculations with a hyperspectral resampling interval set to 2.35 nm, which results in 255 bands over the spectral range between 400 to 1000 nm. The two bands on the upper and lower boundaries of the spectrum are removed, which leads to 253 effective bands.

2. Although the authors added some references (33-38), it would be better to discuss further or compare your results with the published data

Response:

Thanks for your suggestion about further comparison with the published data in the literature. Following your suggestion, we compared the characteristics bands obtained in this paper and those published in the literature. The five characteristic bands extracted in this study are 410 nm, 481 nm, 533 nm, 702 nm, 798 nm. The two bands in 702 nm and 798 nm are are consistent with the 705 nm and 750 nm range used for chlorophyll content retrieval

index in [33]. The characteristic bands of 410 nm, 481 nm, and 533 nm in this paper are similar to those bands used in the study of the rice jointing stage in the literature [34]. Comprehensive analysis shows that there are overlaps and differences in the spectral regions of the characteristic bands among different studies. The differences in characteristic bands are mainly caused by differences in strain varieties, growth periods, environmental conditions, and data processing methods.

The above point is clarified in Section IV of the revised paper.

---

## [Editor Report · Decision Letter 2]

19 Aug 2020

Inversion Modeling of Japonica Rice Canopy

Chlorophyll Content with UAV Hyperspectral

Remote Sensing

PONE-D-20-07987R2

Dear Dr. Xu,

We’re pleased to inform you that your manuscript has been judged scientifically suitable for publication and will be formally accepted for publication once it meets all outstanding technical requirements.

Kind regards,

Wang Li

Academic Editor

PLOS ONE
---

## [Editor Report · Acceptance letter]

24 Aug 2020

PONE-D-20-07987R2 

Inversion Modeling of Japonica Rice Canopy
Chlorophyll Content with UAV Hyperspectral
Remote Sensing 

Dear Dr. Xu:

I'm pleased to inform you that your manuscript has been deemed suitable for publication in PLOS ONE. Congratulations! Your manuscript is now with our production department. 

Kind regards, 

on behalf of

Dr. Wang Li 

Academic Editor

PLOS ONE